# Test Study of the Bridge Cable Corrosion Protection Mechanism Based on Impressed Current Cathodic Protection

Guowen Yao [1,2], Xuanbo He [1,2,*], Jiawei Liu [2], Zengwei Guo [1] and Pengyu Chen [2]

1   State Key Laboratory of Mountain Bridge and Tunnel Engineering, Chongqing Jiaotong University, Chongqing 400074, China
2   School of Civil Engineering, Chongqing Jiaotong University, Chongqing 400074, China
*   Correspondence: 611220080009@mails.cqjtu.edu.cn

**Abstract:** The cable system is an important bearing element of a bridge with stay cables or slings and a matter of major concern in the safety of the bridge structure. Bridge cables are vulnerable to corrosion induced by leakage and soaking during their service life. To solve this problem, and based on the idea of proactive control by means of the impressed current cathodic protection (ICCP) of bridge cables, this study designs and develops an ICCP system device for bridge cable protection. In this study, an accelerated corrosion test was conducted to test the ICCP system of steel wires inside the cables and the cables under acid rain conditions. The corrosion protection behavior of ICCP was analyzed to reveal the corrosion protection mechanism of bridge cable ICCP. The results show that in the cable ICCP system, the impressed current generated by a more negative voltage may improve the efficiency of corrosion protection, but an excessively negative voltage may cause hydrogen embrittlement of the cable steel wire due to overprotection. The rational range of $-1.13$ V to $-1.15$ V was set as the result of the overall consideration. Within this range, the cable is subject to the joint protection of ICCP and sacrificial anode cathodic protection (SACP). Corrosive products can delay the development of cable corrosion to a certain degree; the SACP protection efficiency of the galvanized coat reduces gradually with corrosion development; and cable ICCP protection efficiency increases gradually. The ICCP for cable corrosion protection is transformed from joint protection using both a sacrificial anode and impressed current into protection, mainly using an impressed current.

**Keywords:** bridge engineering; cable steel wire; impressed current cathodic protection; corrosion test

## 1. Introduction

With the increased span length of bridge structures, bridges with cable stays (slings) are becoming indispensably important under the complex engineering environment. The cable system is a key bearing element in bridges with cables or slings. High-performance steel wires within cables demand properties of high strength and corrosion resistance [1,2]. Cables normally have an HDPE sheath to protect the steel wires inside the cable [3–5]. Damage to the HDPE sheath or special sealing protection is usually hard to detect. Furthermore, any replacement of the damaged cables may unavoidably cause traffic interruption on bridges, resulting in serious economic losses [6]. Therefore, cables are often subject to corrosion by leakage and soaking; under the joint action of vehicle-induced vibration [7,8] and wind/rain-induced excitation [9], the cable service life span is greatly shortened [10–12].

Many bridge researchers and scholars have already carried out relevant studies on the corrosion features of cable systems. Yangguang et al. [13] discovered that the uniform corrosion progress of bridge cables can be divided into two stages, namely the corrosion stage of the galvanized coat and the corrosion stage of the iron matrix; such a dramatic reduction in steel wire ductility and other mechanical properties occurs in the stage of iron matrix deterioration [14–16]. Barton [17] and Guowen [18] conducted accelerated

corrosion tests and analyzed the impacts on the mass loss rate, ultimate bearing capacity and percentage elongation caused by corrosion. Nakamura et al. [19] successfully simulated steel wires with different degrees of corrosion and distributed in different spaces by wrapping the steel wires of the main cables of a suspension bridge with a wet gauze. In their study of the electrochemical corrosion rate of galvanized steel wires in bridge cables, Rou and Changqing [20] found that the gradation curves were overall similar with different electrochemical parameters. Zengwei et al. [21] proposed that controlling the $Cl^-$ concentration is an effective way to delay cable corrosion based on cellular automation. Additionally, there are methods for cables that involve more efficient corrosion protection such as coating [22,23] and sealing [24,25].

From the aspect of the corrosion protection of bridges, engineering practices have demonstrated that impressed current cathodic protection (ICCP) systems provide protection to bridges to varying degrees and delay the decay of the bridge elements' mechanical properties [26]. Christodoulou et al. [27] reduced the risk of hydrogen embrittlement of a prestressed concrete structure by applying mixed ICCP to the Tiwai Point Bridge. Rong et al. [28] arranged an ICCP system for the super-long bridge over the Yongding River and controlled the corrosion of the reinforcement in the bridge deck. Green [29] studied an ICCP system for the piers of Lynch's Bridge. All these studies show that ICCP systems provide adequate corrosion protection to the reinforced substructure of bridges. Huyuan [30] conducted real-time monitoring of ICCP in submerged steel tubular piles of the Shanghai Yangtze River Bridge. Dongxiong [31] introduced, in detail, the reinforcement set in the pile cap bottom of the main tower foundation of the Quanzhou Bay Marine Bridge, and the applied ICCP system proved its effectiveness and success in corrosion protection for bridge engineering. Scholars of different disciplines have jointly promoted the technical development of bridge ICCP [32–36]. Using the weight loss method, Bahekar et al. [37] proposed that the corrosion rate of steel bars in reinforced concrete reduces with an increased protective current density. Wei et al. [38] applied ICCP to a reinforced concrete structure under chloride salt conditions and evaluated the ICCP protection efficiency of the reinforcement by measuring the instantaneous breaking potential of the reinforcement and the open circuit potential. Jin-A et al. [39] analyzed specimens of steel bars in reinforced concrete protected jointly with sacrificial anode cathodic protection (SACP) and ICCP and provided a new idea for the corrosion protection of elements in the marine environment.

Currently, a few scholars have applied the idea of ICCP to bridge cables. In this study, we control corrosion development for the cable by proactively applying a current to the cable before corrosion occurs, design and develop a bridge cable ICCP system from the point of view of proactive control with ICCP and evaluate its protection efficiency through tests, analyze its protection mechanism and provide a reference for cable corrosion protection of bridges.

## 2. Cable ICCP Technique

### 2.1. Working Principle of Cable ICCP System

When a corrosive medium invades the interior of a high-density polyethylene (HDPE) sheath of cables in service, corrosive microcells are generated on the steel wire surface of the cables and the galvanized coat and iron matrix of the steel wire will thus be gradually corroded and damaged [40–42]. In accordance with the idea of proactive control with ICCP, an electrical current is manually imposed before corrosion generation to provide electrons to the cable surface based on the electrochemical corrosion principle, with the aid of a power source to inhibit the electron migration generated by the cable itself under corrosion to reduce the redox reaction and, finally, to prevent or delay cable corrosion [43]. The design of the cable ICCP system is shown in Figure 1.

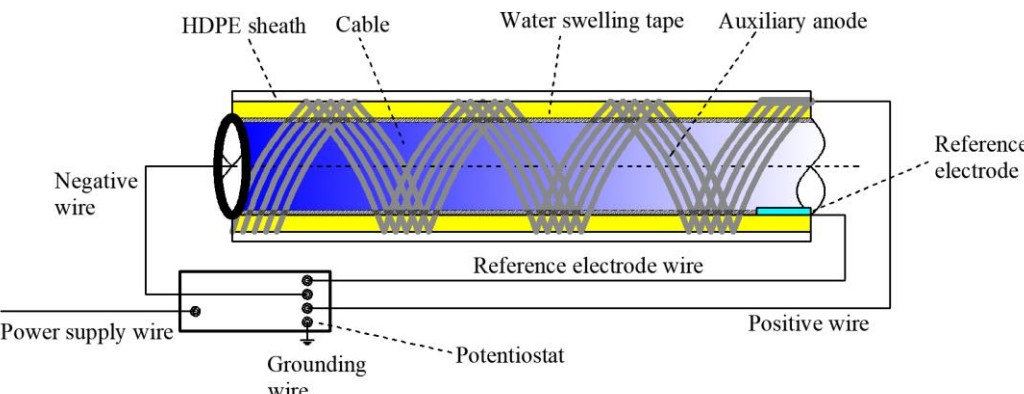

**Figure 1.** Overall design of the cable ICCP system.

The ICCP system for bridge cables applies a water-swelling adhesive tape with high absorption to uniformly and tightly wrap cables when they are in a dry and isolated condition. The stable metal matrix of sound conductivity is selected as the auxiliary anode material. In order to enable the protective current to transmit uniformly and adequately to the cable and its inner steel wires, the geometry of the auxiliary anode is set as a silk shape; the auxiliary anode with a filiform strand or belt/ribbon shape is arranged in parallel, and the cable is twined and wrapped in a spiral manner from the bottom to the top of the external coat of the swelling adhesive tape.

The working principle of the cable ICCP system is that when the cable HDPE sheath is broken or damaged, parts of the corrosion medium will invade the sheath through water–air condensation or rainwater in the service condition. The other parts of the corrosion medium originally existing inside the sheath will be activated by electrolytes and then corrode the cables. At that time, the internal swelling adhesive tape will be soaked gradually. Depending on the moisture level, the swelling adhesive tape will form a conductive body of different degrees and connect with the cable and the auxiliary anode to complete the circuit loop. The cable corrosion will thus be inhibited, owing to the protection of the impressed current. The cable ICCP system and the galvanized coat of the cable steel wire form a joint cathodic protection system consisting of ICCP and SACP, both of which function jointly in achieving improved protection of cables against corrosion. With the evaporation of water collected in the sheath and reduced air moisture, the swelling adhesive tape will dry gradually, and the circuit loop of the cable ICCP will disconnect and no longer protect the cable.

When the power source is connected and circulates the current, the auxiliary anode and cable are connected with the positive and negative terminals of the potentiostat, respectively, and the zero terminal is grounded. In the implementation of cable ICCP, the distance between cables and reference electrode is minimized to reduce the ohmic drop [44] related to the medium, allowing a correct reading of the cable potential.

*2.2. ICCP Corrosion Protection Test*

2.2.1. Simulated Acid Rain Solvent

Cable sections of a cable-stayed bridge were used in the accelerated corrosion test of cable ICCP under an acid rain environment to analyze the feasibility of the cable ICCP system. The test was conducted in accordance with the analysis of measured statistics in the locations vulnerable to severe acid rain pollution in Chongqing, China. The concentration ratios of major components of the acid rain solvent are presented in Table 1 [45].

**Table 1.** Main ion concentration ratios of acid rain solvent.

|  |  | $SO_4^{2-}$ | $NO_3^-$ | $F^-$ | $Cl^-$ | $NH_4^+$ | $Ca^{2+}$ | $Mg^{2+}$ | $Na^+$ | $K^+$ |
|---|---|---|---|---|---|---|---|---|---|---|
| Concentration | Location1 | 7.14 | 3.14 | 0.10 | 0.51 | 1.83 | 1.27 | 0.12 | 0.23 | 0.33 |
| (mg/L) | Location2 | 10.24 | 4.61 | 0.12 | 0.47 | 2.54 | 2.80 | 0.20 | 0.17 | 0.46 |

The acid rain solvent used in the test was prepared using the ion concentration ratios in Table 1. Table 2 lists the main materials required to produce the above solvent. In addition, some essential instruments such as an electronic balance, measuring cylinder, beaker and glass mixing rod were needed, while other technical parameters and amounts are not elaborated.

**Table 2.** Main materials for the solvent preparation.

| Serial Number | Material Name | Specification | Technical Parameter | Quantity |
|---|---|---|---|---|
| 1 | Ammonium sulfate analytically pure | 500 g | AR | 1 bottle |
| 2 | Magnesium sulfate analytically pure | 500 g | AR | 1 bottle |
| 3 | Calcium sulfate analytically pure | 500 g | AR | 1 bottle |
| 4 | Potassium chloride analytically pure | 500 g | AR | 1 bottle |
| 5 | Sodium chloride analytically pure | 500 g | AR | 1 bottle |
| 6 | Diluted sulfuric acid | 500 mL | pH = 1 | 4 bottles |
| 7 | Diluted nitric acid | 500 mL | pH = 1 | 2 bottles |

The procedure for preparing the acid rain solvent is as follows: (1) measure $(NH_4)_2SO_4$ (9.86 g), $MgSO_4$ (2.05 g), $CaSO_4 \cdot 2H_2O$ (12.04 g), NaF (0.27 g) and KCl (0.87 g) with an electronic balance; (2) place them in a 2000 mL beaker, add a suitable amount of distilled water into the beaker and mix them until they are dissolved; (3) place the solvent into a 1#2000 mL volumetric flask to determine the volume; (4) transfer 4 mL of the solvent with a transfer pipe from the 1#2000 mL volumetric flask to the 2000 mL beaker; (5) add a suitable amount of distilled water into the beaker; (6) add pH = 1 diluted sulfuric acid and pH = 1 diluted nitric acid with a volume ratio of 4:1 (keep the ratio 2:1 for $SO_4^{2-}$ to $NO^{3-}$) into the 2000 mL beaker; (7) mix the pH = 3 solvent, and then place it into a 2#2000 mL volumetric flask to determine the volume; (8) transfer the pH = 3 solvent from the volumetric flask to a measuring cylinder, thus completing the preparation of the acid rain solvent.

A similar approach is applied to the preparation of acid rain solvent with pH = 5 and neutral solvent with pH = 7 at an amount of 2000 mL.

2.2.2. Auxiliary Anode and Reference Electrode

The common auxiliary anode materials are lead silver alloy, platinum, platinum-coated metal, etc. Metal lead is not only harmful to human health but may also pollute the environment. It is generally used in high-chloride-ion media and thus obviously has limited application in cable ICCP systems. Platinum, as a high-valued metal [46], has stable electrochemical properties and higher corrosion resistance, but it has a higher density and price. Therefore, extensive application of platinum can be difficult. Metal platinum is normally used as the coating material of auxiliary anodes, and it is also used jointly with metal titanium, which has the same electrochemical properties. Platinum plating on a titanium anode can not only satisfy the basic requirement of the auxiliary anode material but also bear a higher current with a relatively lower cost. The final choice for the cable ICCP system was platinum plating on a titanium anode as the auxiliary material.

The ICCP protection efficiency of the cables is judged and evaluated based on the potential difference obtained by measuring the reference electrode. The principle is that the potential of the electrode to be measured can be obtained by measuring the electrodynamic force of the battery, which is formed by the obtained reference electrode with a known electrode potential and the electrode to be measured. The saturated calomel reference electrode has a longer service life and higher stability, strength and corrosion resistance, so

a saturated calomel reference electrode was finally chosen to be the reference electrode for the cable ICCP system.

### 2.2.3. Protective Potential

Before the impressed current is imposed on the cables, the appropriate range of the protective potential should be tentatively selected based on the essential principle of cathodic protection to avoid excessive cable protection, hydrogen embrittlement or unnecessary energy consumption after the prescribed cable protection effect has been achieved.

The steel tendon inside the cable is made of PES (C) 5–85 galvanized parallel steel wires. The steel wires' diameter $\varphi = 5.0$ mm and the number of steel wires in the cable s = 85, with a tensile strength of $f_{pk} = 1770$ MPa. The electrochemical test uses a three-level system and the measurement steps are as follows: (1) place the cable, reference electrode and platinum-plated titanium anode into the insulated column container and add the acid rain solution configured in advance; after the cable has been immersed in the acid rain solution for 2 h, turn on the power switch of the constant potential meter, allow the instrument to warm up for 30 min and then set the constant potential meter to the constant potential working state. (2) Connect the working electrode of the constant potential meter to the cable, the reference electrode to the saturated glycerol reference electrode and the counter electrode to the platinum-plated titanium auxiliary anode, and press the *load* key and the *reference* key at the same time to make the instrument work in the reference state. The *reference* indicator lights up and the value shown in the potential display bar is the reference potential of the cable in the acid rain solution relative to the reference electrode, which is the open circuit potential of the cable. (3) Press the *load* and the *electrolytic cell* buttons at the same time to place the instrument in the electrolytic cell mode, with the open circuit potential of the cable as the initial value, adjust the voltage by rotating the *DC* button to the positive direction by 20 mV each time, and record and observe the corresponding current and voltage data for one minute after each voltage adjustment. (4) Taking the open circuit potential of the cable as the initial value, adjust the voltage by rotating the *DC* button in a negative direction by 20 mV each time, and record and observe the corresponding current and voltage data for one minute after each voltage adjustment. (5) Record the measured voltage and current data and draw a polarization curve. Repeat the above steps to measure the polarization curves of the cable in different pH solutions, as shown in Figure 2.

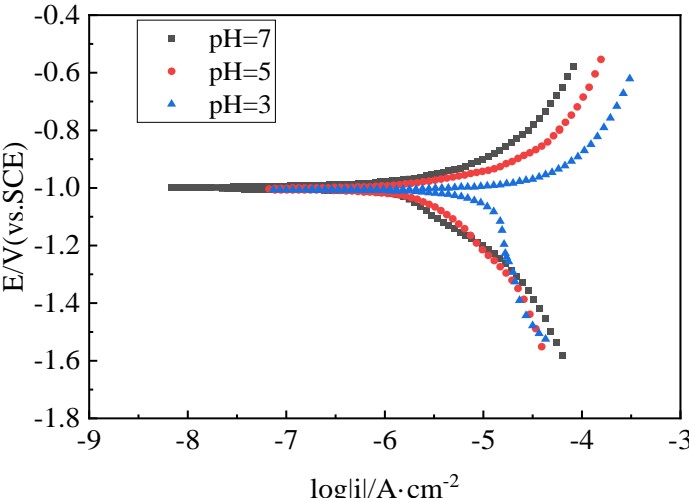

**Figure 2.** Cable polarization curve.

It is worth noting that when corrosion of a specimen occurs in an acidic solution, an ambient temperature of about 35 °C will accelerate its corrosion rate [47,48]. Therefore, in order to accelerate the efficiency of the test and verify the feasibility of the ICCP system,

the test was conducted in the environmental test chamber of the State Key Laboratory of Mountain Bridge and Tunnel Engineering, Chongqing Jiaotong University, with the ambient temperature controlled at 35 °C and the temperature error controlled within 2 °C.

The polarization curves of the cables in acid rain solvents with pH = 3, pH = 5 and neutral solvent with pH = 7 were measured using a static method and a potentiostat, and the curves are shown in Figure 2. The polarization curves show that the open circuit potentials were −1.006 V (vs. SCE), −1.001 V (vs. SCE) and −0.998 V (vs. SCE), respectively, and the difference among the three potentials was controlled within 10 mV. The study of the corrosion protection test of the cable in an acid rain solvent with pH = 3 is also suitable for solvents with pH = 5 and pH = 7. The cable in the solvent with a lower pH displayed an accelerated corrosion speed; hence, the selected acid rain solvent with pH = 3 can improve the efficiency of the ICCP corrosion protection test and highlight the results of the test.

The protective potential was selected according to the polarization curve of the cable in the acid rain solvent with pH = 3. The protective potential output range is generally taken to be the range between the two inflection points of the polarization curve. The first inflection point is generally taken as the minimum protective unit, approximately −1.070 V, and a value that is more positive than this potential will not achieve the expected protection effect. The second inflection point is taken as the maximum protective potential, approximately −1.400 V. During the measurement, the protective current increases gradually as its voltage becomes negative; meanwhile, a few hydrogen bubbles are generated on the cable surface. When the protection voltage was set more negative than −1.130 V, the surface of the cable showed obvious hydrogen bubbles, which shows that this potential is susceptible to the hydrogen evolution reaction and hydrogen embrittlement is likely to occur. When the protection voltage of the cable ICCP system is more negative than −1.400 V, many hydrogen bubbles are generated on the cable surface, which presumes that the hydrogen embrittlement is extremely likely to occur near this protection voltage, which endangers the safety of the cable in service. As a result, the maximum protective potential of the organic zinc-rich coating was −1.300 V, and the protective potential output range of the cable under test was tentatively set in the range between −1.070 V and −1.300 V.

### 2.2.4. Cable ICCP System

An industrial cutting machine was applied to cut and strip the HDPE sheath in the pre-treatment, as shown in Figure 3a. Three sections of cable were cut lengthwise, with each section having a length of 0.6 m for test purposes. Cable corrosion was simulated on the three cable sections under different impressed currents. A 0.2 m long sheath close to the ends of the cable was reserved, which always maintained the original adhesion to the cable when stripping the HDPE sheath off the cable to retain the integrity of the steel wires inside the cable.

The preparation of the three cable sections for the test is shown in Figure 3b. Copper wire was twined around the naked cable at end A and served as the conducting wire. A dry, high-quality sponge was selected to wrap the naked cable at end B and served as the swelling adhesive tape. Strands of belts partially coated with platinum and titanium were twined around the external layer of the swelling adhesive belt and served as the auxiliary anode.

After the cable ICCP system was completed and installed, as shown in Figure 3c,d, end B of the cable section wrapped in a platinum–titanium-coated belt/ribbon was tilted downward and dipped into the three isolated cylindrical vessels containing the acid rain solvent. According to the design in Figure 1, the lead wire was connected to the positive terminal of the potentiostat below the coated belt/ribbon in close proximity to the cable surface, and the lead copper wire at the end of the cable was connected to the negative terminal of the potentiostat. The calomel reference electrode was fixed below the reserved HDPE sheath, in close proximity to the cable surface and connected to the terminal of the reference electrode of the potentiostat. Finally, the external power sources for the three cables on the right in Figure 3d were connected to carry out the ICCP test. The cables on

the left in Figure 3d were pre-treated by cutting and stripping the HDPE without the cable ICCP system, and the naked cable was dipped into the vessel containing the acid solvent and then placed under a natural corrosion condition to form a contrast to the other three cables.

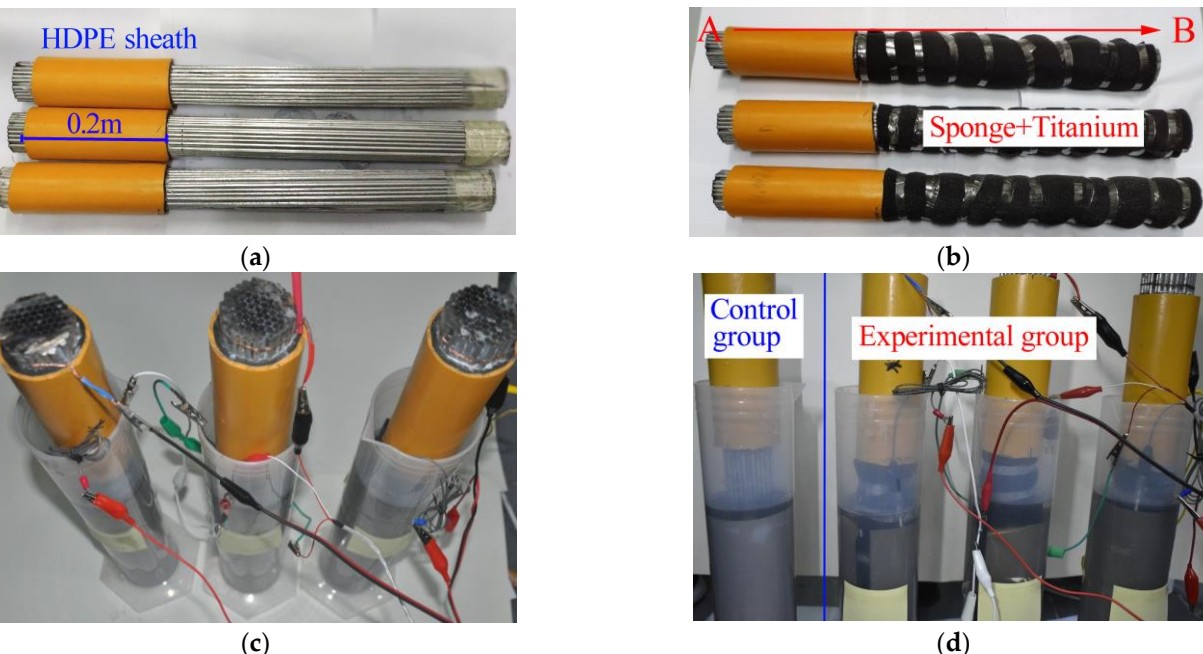

**Figure 3.** Cable ICCP corrosion test: (**a**) cable pre-treatment; (**b**) cable preparation; (**c**) cable ICCP system preparation; (**d**) corrosion control group and test group.

## 3. Results and Analysis

### 3.1. Analysis of Surface Configuration of Corroded Cables

Although the more negative protection voltage may have the more effective corrosion protection effect on the cable, hydrogen embrittlement may occur, so a more positive protection voltage than −1.130 V was set for one group of cables, and a more negative protection voltage than −1.130 V was set for other two groups of cables. Protective voltages of −1.070 V, −1.150 V and −1.250 V were set for the cable ICCP system of the three tests according to the tentatively selected protective potential outputs, so that the corresponding protection currents flowed in the cables. The cable in the comparison test group was not equipped with an impressed current and was placed under a natural corrosion condition. Since the pH of the acid rain solvent changed in the test process, in order to ensure that the acid rain solvent always maintained a state of pH = 3, the pH of acid rain solvent in the different vessels was checked with an electronic pH tester at intervals of 6 h. Then, a diluent was prepared with diluted sulfuric acid of pH = 1 and diluted nitric acid of pH = 1 as per the volume ratio of 4:1, and the diluted solvent was used to regulate the pH value in each vessel to reach pH 3. To avoid the regulated $SO_4^{2-}$ and $NO_3^-$ accumulation, which can generate a test error, the acid rain solvent in the vessels was disposed of at intervals of 48 h, and the acid rain solution with pH = 3 was reconfigured. By doing this, the cable was always immersed in acid rain with a high concentration in a state of corrosion acceleration. The corrosion protection of bridge cables, in particular, is subject to the impacts of the dissipation of the galvanized coating and iron matrix's preliminary participation in the reaction, which is the initial stage of cable corrosion development. In order to focus on the early behavior of cable corrosion development, the corrosion test time was set to 336 h.

After the end of the corrosion protection test of bridge cables, the sponge wrapped around the surface of the corrosion cable and its platinum–titanium-coated belt was removed, the cable surface was cleaned with distilled water and the surface conditions of the

four groups of cables were observed and recorded. The cable without voltage protection suffered serious corrosion, as shown in Figure 4.

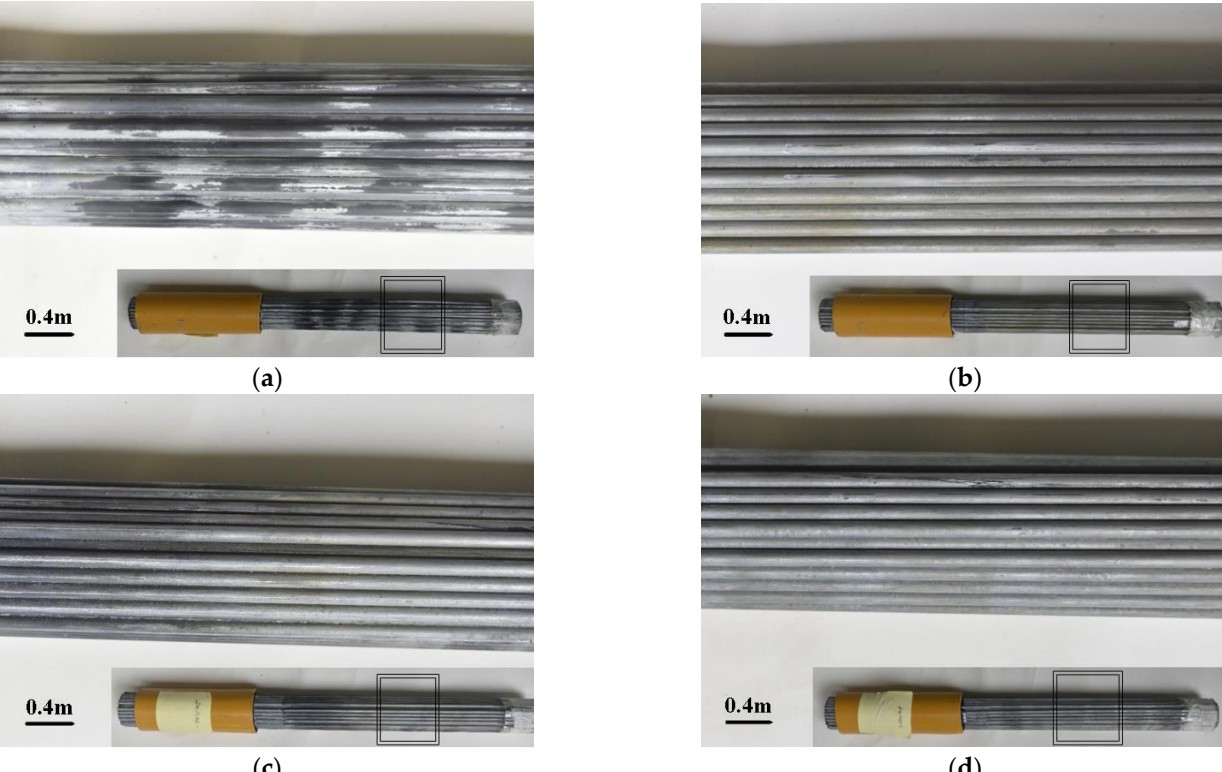

**Figure 4.** Surface appearance of corroded cable: (**a**) without voltage protection; (**b**) protected by a voltage of −1.070 V; (**c**) protected by a voltage of −1.150 V; (**d**) protected by a voltage of −1.250 V.

The local galvanized coating of the steel wires was completely damaged due to the corrosion action of the acid rain solvent, and at locations with a darker color, the iron matrix was clearly visible. With a protection voltage of −1.070 V, a light-yellow cover was generated on the cable surface, and the galvanized cover of the steel wires was slightly damaged. With a protection voltage of −1.150 V, the cable surface appearance was similar to that of the cable protected by a voltage of −1.070 V. The light-yellow cover generated on the cable surface was distributed sparsely, and the damage of the galvanized cover of the steel wires was at the same level as the damage to the cable protected by a voltage of −1.070 V. The light-yellow cover generated on the cable surface protected by a voltage of −1.250 V was densely and uniformly distributed, and the galvanized coat of the steel wires was relatively complete.

### 3.2. Close Observation and Weight Loss Analysis of Corroded Steel Wire

A visual analysis of the cable corrosion protected by different voltages was carried out to determine the different roles of ICCP in inhibiting the corrosion of the cable. However, due to the large quantity of steel wires in the cable sections, and the varied distribution of the steel wires inside the cable, it is hard to digitally treat variables of different types. In order to quantify the corrosion protection efficiency of the cable ICCP system, the ICCP of single steel wires under acid rain conditions was tested after the cable corrosion tests had already been completed.

In the test, a denser change interval of the protective voltage was set for the steel wires, and the test steel wire groups consisting of four steel wires were protected by voltages of −1.070 V, −1.130 V, −1.200 V and −1.300 V. Two steel wires without voltage protection formed the comparison test group subjected to corrosion for durations of 168 h and 336 h to analyze the impact of the corrosion duration. The steel wires used in the test were

galvanized steel wires of the same type as those in the cable. Each steel wire was cut to a length of 0.5 m. Before the test, each steel wire was surface-polished with waterproof sandpaper to produce a smooth surface, cleaned with distilled water and then dried in an oven. Then, the same steel wire was cleaned and washed with anhydrous ethanol before being placed in the dryer until further weighing. After being dried, the six steel wires were weighed with an electronic scale, and their respective weights were recorded. The following test used identical procedures to the cable ICCP test under acid rain conditions.

The duration of test stage 2 was 336 h. The steel wires were cleaned with distilled water, and when they were dry, a digital microscope was used to observe the micro appearance of the steel wires, as shown in Figure 5.

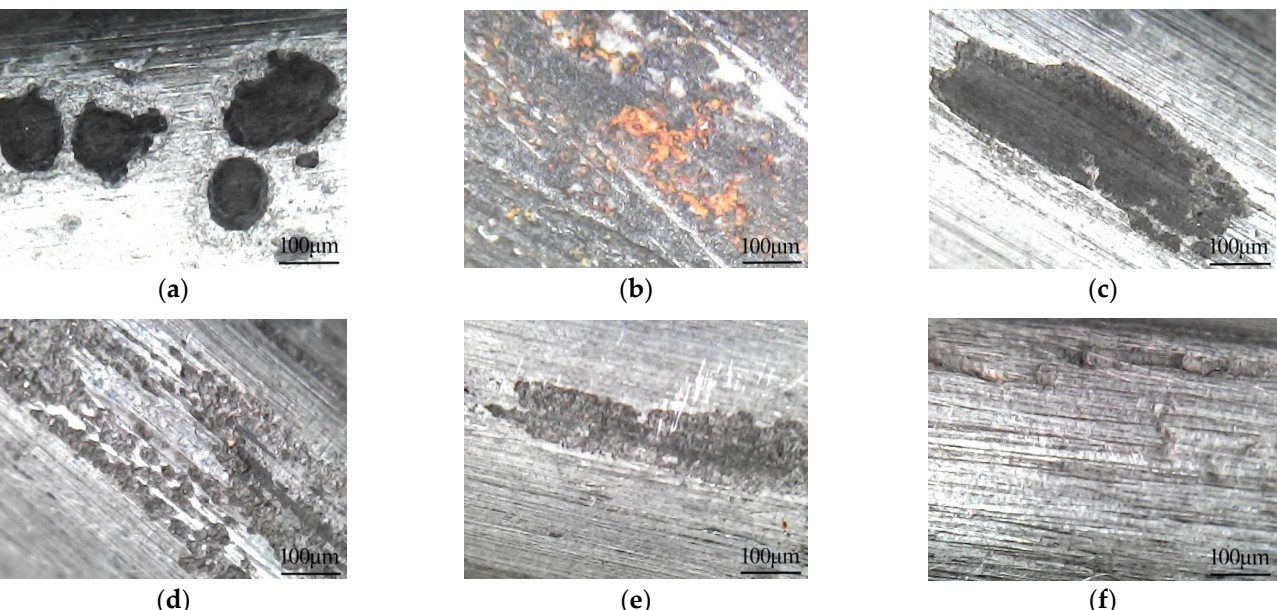

**Figure 5.** Micro appearance of the corroded steel wire: (**a**) without voltage protection (118 h); (**b**) without voltage protection; (**c**) protected by a voltage of −1.070 V; (**d**) protected by a voltage of −1.130 V; (**e**) protected by a voltage of −1.200 V; (**f**) protected by a voltage of −1.300 V.

The galvanized coat of the steel wire without voltage protection was seriously damaged after being immersed in the acid rain solvent with pH = 3 for a duration of 168 h. Many etch pits a millimeter in scale appeared on the steel wire surface. Due to the exposed iron matrix, sunken black spots were densely distributed on the steel wire surface and red-brown rust was generated on the iron matrix. By that time, the steel wire was still being well-protected by the galvanized coat against corrosion.

When the corrosion duration reached 336 h, the galvanized coat of the steel wire without voltage protection was almost completely damaged, and the corrosion medium's corroded iron matrix and red-brown rust were generated on the steel wire surface. By contrast, the galvanized coat of the steel wire protected by a voltage of −1.070 V was locally damaged at a shallow depth, and the iron matrix was exposed at the damage location but without red-brown rust; the corrosion damage was smaller both in intensity and range, compared with the steel wires without voltage protection. By that time, the galvanized coat was still protecting the steel wire with SACP. Through comprehensive analysis, the steel wires protected by voltages of −1.130 V and −1.200 V suffered from slightly less corrosion than the steel wire protected by a voltage of −1.070 V. The galvanized coat of the steel wire protected by a voltage of −1.300 V suffered the least damage. Hence, the cable ICCP system can efficiently reduce the damage to the galvanized coat of steel wires and prevent corrosion damage to the iron matrix.

Furthermore, when the test reached 168 h, one corroded steel wire without voltage protection (without removing the rust coat) was withdrawn and placed in a dryer; then,

after being dried, its weight was measured with an electronic balance. When the test reached 336 h, the other corroded steel wire without voltage protection was withdrawn along with the four steel wires protected by a voltage (without removing the iron rust coat), which were then placed in a dryer until they were dry; following this, they were placed on an electronic scale to measure and record their weight. Then, the surface rust coat on each steel wire was cleaned and removed with a brush using flowing water, and the cleaned steel wires were placed in a dryer until they dried. Their weight was measured and recorded with an electronic balance, and then the mean corrosion rate of each steel wire during the test was calculated using Equation (1) according to the weight loss from steel wire corrosion.

$$v = \frac{M_1 - M_2}{St} \tag{1}$$

where $v$ is the mean corrosion rate of the steel wire, $g \cdot m^{-2} \cdot h^{-1}$; $M_1$ is the mass per unit length of the steel wire before corrosion, $g \cdot m^{-1}$; $M_2$ is the mass per unit length of the steel wire after corrosion, $g \cdot m^{-1}$; $S$ is the effective surface area of the steel wire exposed to corrosion, $m^2$; $t$ is the corrosion time of the steel wire, h.

The weights and calculations are shown in Table 3.

**Table 3.** Steel wire weight loss from corrosion, and corrosion rate.

| Corrossion Time (h) | Group | Mass before Corrosion (g) | Mass after Corrosion (g) | Weight Loss Quantity (g) | Mean Corrosion Rate ($g \cdot m^{-2} \cdot h^{-1}$) |
|---|---|---|---|---|---|
| 168 | Without voltage protection (rust not removed) | 75.32 | 74.88 | 0.44 | 0.3335 |
| 336 | Without voltage protection (rust not removed) | 75.32 | 74.09 | 1.23 | 0.4661 |
| | Without voltage protection | 75.32 | 73.66 | 1.66 | 0.6290 |
| | With protection of −1.070 V | 75.10 | 74.96 | 0.14 | 0.0531 |
| | With protection of −1.130 V | 75.11 | 75.05 | 0.06 | 0.0227 |
| | With protection of −1.200 V | 75.27 | 75.22 | 0.05 | 0.0189 |
| | With protection of −1.300 V | 75.18 | 75.15 | 0.03 | 0.0114 |

Table 3 clearly shows that the weight loss of the steel wire without voltage protection was the greatest, and the mean corrosion rate was the fastest. When the steel wire was protected by an impressed current, its corrosion rate was noticeably reduced. The more negative the protection voltage set for the steel wire, the greater the corrosion-inhibiting effect of the cable ICCP. The contrast analysis of the mean corrosion rate at different times of the steel wire without removal of rust showed that the mean corrosion rates of both at their corresponding test durations were basically identical. The mean corrosion rate of the steel wire corroded for 336 h was slightly more than that of the steel wire with a corrosion duration of 168 h, the mean corrosion rate of the steel wire shows that the development of steel wire corrosion without voltage protection is increasingly rapid.

*3.3. Fracture Damage Analysis of Corroded Steel Wire*

A computer-controlled twin-arm-type electronic universal testing machine equipped with the test analysis software SANS-Power Test DOOC from the structural laboratory of Chongqing Jiaotong University was used to test the mechanical properties of the corroded steel wires. Equipment model: WDW-50; equipment specifications: 5 t; equipment accuracy level: 0.5 level. The equipment can complete the tensile, compression, bending, shear and basic static mechanical tests of composite materials as well as tear, friction and cycle tests. In order to further determine and set the protective range of the voltage in the ICCP system for the steel wires, the four corroded steel wires protected by different voltages were uniaxially tension-tested with the universal testing machine, and the stretching loading rate was 3 mm/min. The steel wire was tensioned until it broke, determining whether the steel wire suffered from hydrogen embrittlement due to a low protection potential setting

by stretching the steel wire fracture shape, and thus determining the maximum protection voltage for the steel wire, as shown in Figure 6.

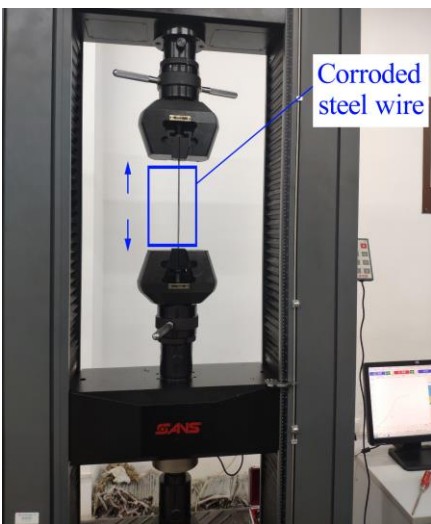

**Figure 6.** Stretching test with universal testing machine.

The steel wire was stretched by the universal testing machine until it broke. The tensile strength, strength loss rate and elongation rate after break of the four corroded steel wires are shown in Table 4. The fracture surface appearance and shape of the respective steel wire were recorded, as shown in Figure 7, of which the steel wire tensile strength was recorded by the stretching equipment of the universal testing machine. Its strength loss rate was calculated using Equation (2), and the elongation rate after fracture was calculated using Equation (3).

$$p = \frac{f_{pk} - f_a}{f_{pk}} \times 100\% \tag{2}$$

where $p$ is the strength loss rate of the corroded steel wire, %; $f_{pk}$ is the initial tensile strength before steel wire corrosion, MPa; $f_a$ is the tensile strength after steel wire corrosion, MPa.

$$w = \frac{(l_a - 2l) - (l_b - 2l)}{l_b - 2l} \times 100\% \tag{3}$$

where $w$ is the elongation rate after fracture of the corroded steel wire, %; $l$ is the length reduction of the steel wire end to be fixed to the universal testing machine, m; $l_a$ is the total length of the two sections of fractured steel wires, m; and $l_b$ is the initial steel wire length before its fracture, m.

**Table 4.** Mechanical indices of the fracturing of corroded steel wire.

|  | −1.070 V Protection | −1.130 V Protection | −1.200 V Protection | −1.300 V Protection |
|---|---|---|---|---|
| Tensile strength (MPa) | 1648 | 1693 | 1586 | 1572 |
| Strength loss rate (%) | 6.89 | 4.35 | 10.40 | 11.19 |
| Elongation rate after fracture (%) | 4.19 | 5.06 | 3.23 | 2.95 |

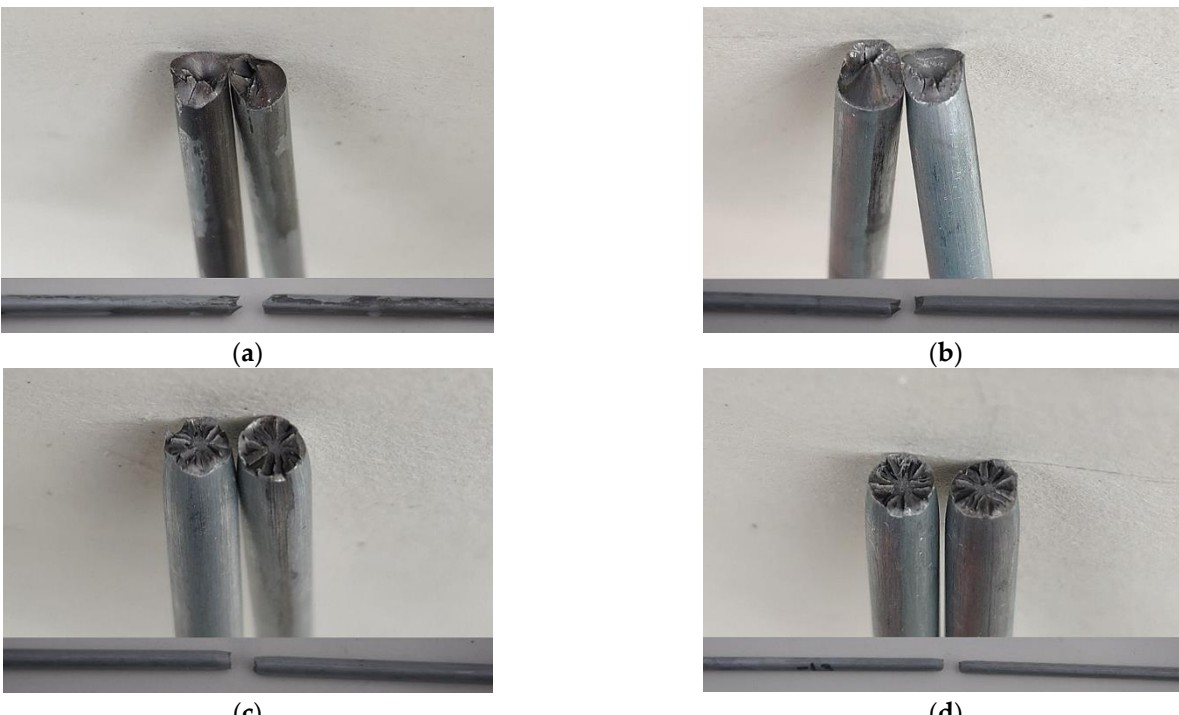

**Figure 7.** Fracture surface appearance of corroded steel wire: (**a**) −1.070 V protection; (**b**) −1.130 V protection; (**c**) −1.200 V protection; (**d**) −1.300 V protection.

Whether the steel wire undergoes hydrogen embrittlement due to overprotection can be judged by the steel wire fracture mechanical index and its fracture surface appearance feature. Table 4 shows that when the voltage changed from −1.070 V to −1.113 V, the more negative protection voltage better protected the steel wire in terms of its tensile strength, and the steel wire's greatest tensile strength was generated when the steel wire was protected by −1.130 V, with a strength loss rate of only 4.35%. The strength loss rates of the steel wires protected by the other three voltages exceeded 5% without exception. The strength loss rate of the steel wires protected by voltages of −1.200 V and −1.300 V even exceeded 10%.

The micro appearance of the steel wires protected by different voltages shown in Figure 5 and the corrosion weight loss and corrosion rate of the steel wires were analyzed. The result shows that although a better protection effect is reflected in the micro appearance, weight loss and mean corrosion rate of the steel wire protected by a more negative voltage, a protection voltage more negative than −1.130 V may lead to a reduced steel wire tensile strength, and its elongation rate after its fracture may be considerably reduced compared with a steel wire protected by a more positive voltage after steel wire fracture. The post-fracture elongation rate of the steel wire protected by voltages of −1.200 V and −1.300 V was 3.23% and 2.95%, compared with the post-fracture elongation rate of 5.06% of the steel wire protected by a voltage of −1.130 V. The ductility of the steel wire was noticeably reduced, and therefore it can be tentatively judged that a protection voltage more negative than −1.200 V may cause hydrogen embrittlement of the steel wire.

Figure 7a,b show that the fracture of the steel wires protected by voltages of −1.070 V and −1.130 V appears to be a pure shear fracture surface, which is crude and bears an obvious shear ridge, and the fracture direction has an angle of about 45 degrees in the stretching direction.

The fracture demonstrates slight necking down, and the fracture surface shows cracks developed from its interior due to the tolerant fracture. Being different, the steel wires protected by −1.200 V and −1.300 V have flatter, radial fractures, and the steel wire protected by a voltage of −1.300 V shows a more obvious radial ridge but with little plastic

deformation, which is in line with brittle fracture characteristics. By further combining with the fracture mechanical index of the steel wires in Table 4 and the appearance of the fractures of all steel wires, it can be judged that the steel wires protected by voltages of −1.070 V and −1.130 V are subject to a lower possibility of hydrogen embrittlement, while the steel wires protected by voltages of −1.200 V and −1.300 V are subject to a higher possibility of hydrogen embrittlement damage, where the steel wire hydrogen embrittlement increases continually with the more negative voltage. This phenomenon is basically consistent with the pattern of polarization curves in Figure 2. Hence, in the application of the ICCP system to steel wire corrosion protection, it is important to prevent the occurrence of hydrogen embrittlement while protecting the steel wire from corrosion. From the overall consideration of the corrosion development of steel wires inside the cables and their fracture, the protection voltage should be controlled within a reasonable range between −1.130 V and −1.150 V, within which the impressed current effectively inhibits the corrosion development of cables and prevents hydrogen embrittlement.

## 4. Protection Mechanism of Cable ICCP System

In the cable ICCP, the internal mechanism of corrosion protection is the joint protection from the cable ICCP system and SACP of the galvanized coat. The SACP uses zinc as the galvanized coat, which has a lower potential than the cable steel wire to provide a protection current to the iron matrix. It uses the elemental zinc coating as the anode and protects the cathode surface using the current generated by the dissolution of the anode. Another function of the galvanized coat is to provide a physical barrier layer for the iron matrix. The demand of the cable for the protection current is further reduced due to the barrier layer, which facilitates the uniform transmission of the protection current from the cable surface to its interior. In the corrosion development of SACP, the open circuit of the galvanized steel structure changes gradually from the corrosion potential of the zinc electrode to that of the iron electrode [49]. A potentiostat was used to measure the potential changes in the open circuit of the steel wires inside the cable before and after corrosion, and the cable protected by different voltages. The open circuit changes are shown in Figure 8.

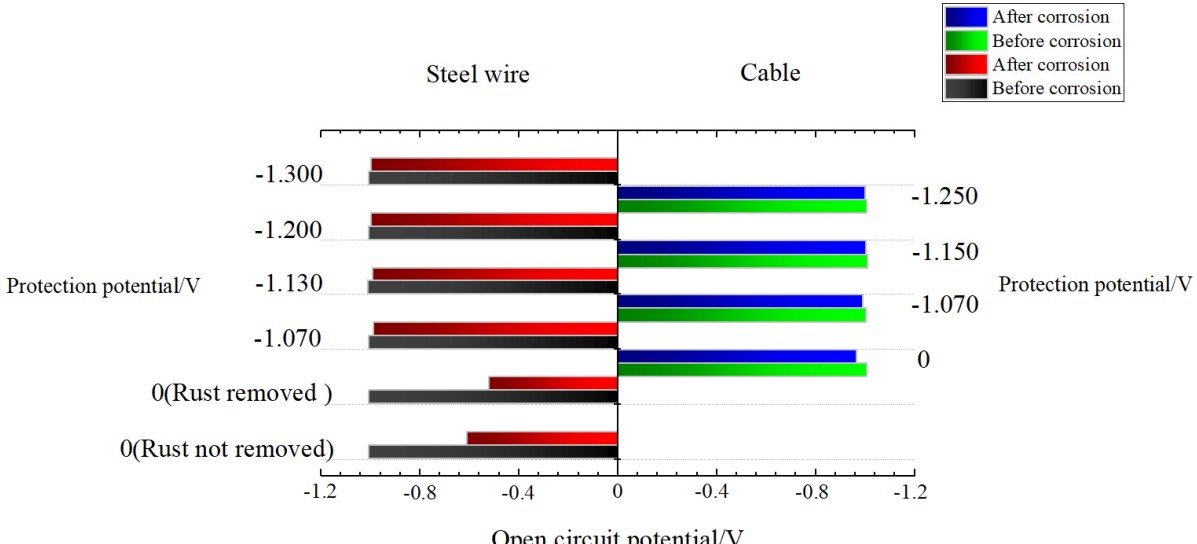

**Figure 8.** Open circuit potential of the cable and the steel wires inside the cable.

Through analysis of Figure 8, it is concluded that the open circuit potential changes of the steel wires inside the cable and the cable in the ICCP system also meet this trend, and their open circuit potential changes to steadily become more positive with the corrosion development; the change range presents a positive correlation with the corrosion degree. The potential of the open circuit of the cable before corrosion without voltage protection in

acid rain solution was −1.006 V, and after 336 h corrosion, the open circuit potential was −0.965 V, with an increment of 0.041 V. The open circuit potential of the cable protected by a voltage of −1.070 V before and after corrosion presented the most obvious changes among the cables protected by different voltages, with an increment of 0.014 V. In identical conditions, the open circuit potential of the corroded single steel wire increased and was even 0.020 V higher than that before corrosion. Under the premise of constant control of the protection potential, this change trend may increase the potential difference of the working electrode in the ICCP system, causing an increased impressed current density within the control range.

The following analysis of this phenomenon was conducted: The galvanized coat on the steel wire surface may deteriorate and break with corrosion development. Because the sacrificial anode metal is worn out, the SACP protection efficiency of the galvanized coat gradually reduces, while the ICCP protection efficiency increases steadily. The cathodic protection against cable corrosion is transformed from the concerted control of the sacrificial anode and impressed current to the control dominated by the impressed current, which inhibits corrosion generation. The working mechanism is shown in Figure 9. In Figure 9, direction 1 is the electronic transmission path of SACP in the cable corrosion process; direction 2 means that the cable ICCP system, as a corrosion protection device, provides cable corrosion protection; direction 3 is the corrosion of the iron matrix of the cable. The electron transmission of direction 1 and direction 3 is driven by the potential difference between different metals, while the electron transmission of direction 2 is driven by the impressed current. This phenomenon that the contribution of electrons transmitted by different paths changes with corrosion development also provides another way of thinking about the setting of the protection voltage in ICCP systems: the protection voltage can be further controlled and reduced in the range between −1.130 V and −1.150 V to avoid overprotection of the cable and an excessively large protection current. This is key to preventing cable hydrogen embrittlement.

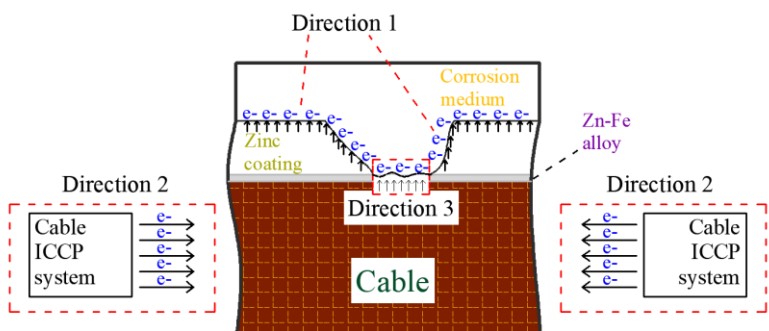

**Figure 9.** Mechanism of joint cathodic protection.

In addition, through contrast analysis of the open circuit potentials of the steel wires with and without removal of rust after being corroded naturally, as shown in Figure 8, it can be concluded that the open circuit potential of the steel wires with rust removal reached −0.612 V, while the open circuit potential of the steel wires without rust removal reached only −0.521 V, which obviously showed a more positive open circuit potential, by which a smaller potential difference was generated, which is more advantageous to the corrosion protection of the cable itself. This result indicates that the common encapsulation and accumulation of both galvanized layer corrosion products in the SACP stage and iron matrix corrosion products in the ICCP stage can delay corrosion development to a certain degree.

## 5. Conclusions

A cable ICCP system was designed and developed based on the idea of proactive control of ICCP to simulate and test the accelerated corrosion of a cable under simulated

acid rain conditions of the ICCP system. This study analyzed the behavior of cable ICCP in its corrosion protection of a cable and revealed the corrosion protection mechanism of cable ICCP for bridges. The following conclusions are drawn:

The impressed current generated by a more negative protection voltage can provide more efficient corrosion protection to bridge cables. However, an excessively negative voltage may lead to hydrogen embrittlement of the steel wires inside the cable due to overprotection. The test showed that the rational range of the protection voltage is between −1.130 V and −1.150 V. When the protection voltage output by ICCP is more negative than the set range, the corroded steel wire will undergo brittle fracture and the fracture surface will have a radial shape. When the protection voltage output by ICCP is more positive than the set range, the corroded steel wire will undergo a tolerant fracture, showing necking down and a shear fracture ridge. The bridge cables are jointly protected by both the impressed current and cathodic protection of the sacrificial anode. The corrosion product can delay the corrosion development of the cables. The protection efficiency of the cable ICCP system increases gradually with corrosion development, and the SACP protection efficiency of the galvanized coat decreases gradually with corrosion development. The cable corrosion protection is transformed from joint protection to protection dominated by the impressed current, thus steadily inhibiting corrosion generation.

**Author Contributions:** Conceptualization, G.Y., X.H. and Z.G.; methodology, X.H.; software, J.L., Z.G. and P.C.; validation, G.Y., J.L. and P.C.; formal analysis, G.Y., X.H., Z.G. and P.C.; investigation, X.H.; resources, J.L., Z.G. and P.C.; data curation, G.Y., X.H., Z.G. and J.L.; writing—original draft preparation, G.Y. and X.H.; writing—review and editing, G.Y. and X.H.; visualization, G.Y. and X.H.; supervision, G.Y. and X.H. All authors have read and agreed to the published version of the manuscript.

**Funding:** The National Natural Science Foundation of China (Grant No. 52178273), the National Natural Science Foundation of China (Grant No. 51878106), the Natural Science Foundation of Chongqing (Grant No. cstc2021jcyj-msxmX1159), the Chongqing Talent Plan Project (Grant No. cstc2022ycjh-bgzxm0124), the Open Fund Project of State Key Laboratory of Mountain Bridge and Tunnel Engineering (Grant No. SKLBT-YF2105), the Chongqing Project of Joint Training Base Construction for Postgraduates (Grant No. JDLHPYJD2020004).

**Data Availability Statement:** The data presented in this study are available on request from the corresponding author.

**Conflicts of Interest:** The authors declare no conflict of interest.

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
