# Peer review of "Test Study of the Bridge Cable Corrosion Protection Mechanism Based on Impressed Current Cathodic Protection"

_lubricants, doi:10.3390/lubricants11010030_

Round 1

Reviewer 1 Report

This manuscript deals with en experimental examination of the cathodic protection system (a combination of impressed current  and sacrificial anode systems) applied to bridge cables after the external HDPE sheath is broken and aqueous electrolyte invades the internal volume interstices.  The examination consisted in electrochemical analyses of OCP and linear voltammetry, combined with SEM morphology and mechanical testing of corroded samples.

 The works is novel as the corrosion protection system in an interesting method to be applied not only to bridge cables but also to other cables exposed to seawater and underground conditions.  

 Unfortunately the manuscript is hard to read. There is many grammatical errors with badly structures sentences that makes this manuscript unsuitable to be published in the current form.

I recommend a mayor revision to upgrade the manuscript in order to clarify many sentences and  ideas.

 I attached a pdf manuscript with popup remarks to help in the upgrading work of the manuscript.  

Author Response

Response to Reviewer 1 Comments

Dear reviewers and editors of Lubricants, thank you for reviewing this paper and providing valuable comments during your busy schedule. Our team further optimized the structure of the manuscript, sorted out the conclusions again, and improved the language. We have carefully read and considered the issues and suggestions you pointed out. After discussing and reorganizing them together, we now explain and correct them.

Thank you for the detailed pdf manuscript, which makes it very clear for us to recognize our mistakes.

Point 1: The part about the revision and improvement of language expression.

Response 1: Thank you for your detailed comments on language errors in the manuscript. We have upgraded and rewritten the expressions of all the sentences you marked as suggested for revision in the original manuscript, corrected words and phrases, including singular and plural, initial capitalization, incorrect phrase expressions, replacement of words and missing conjunctions, and asked professionals to touch up the entire manuscript. The revised manuscript has been uploaded, and the revised sections have been highlighted in yellow for your review.

Point 2: “The auxiliary anode material shall be of sound conductivity and stability”- You mean low electrical resistance and chemical stability?

Response 2: Your suggestion is correct, we want to express low electrical resistance and chemical stability, and now we have corrected this sentence.

Delete: The auxiliary anode material shall be of sound conductivity and stability.

Replace with: The auxiliary anode material should have low electrical resistance and chemical stability.

Point 3: “amount of s=85nos”- What is this?

Response 3: Combined with the previous description, we wanted to convey the idea that this section of the cable was made up of 85 steel wires. The expression in the original manuscript was poorly expressed, and we have now modified it.

Delete: The steel tendon inside the cable is made of PES (C) 5-85 galvanized parallel steel wires. The steel wire is of diameter φ=5.0mm, amount of s=85nos with tensile strength of fpk=1770MPa.

Replace with: The steel tendon inside the cable is made of PES (C) 5-85 galvanized parallel steel wires. The steel wires’ diameter φ=5.0mm, and the number of steel wires in the cable s=85, with a tensile strength of fpk=1770MPa.

Point 4: “static method”- What is this?

Response 4: What the static method is trying to convey is that keep the electrode potential at a certain value, measure the corresponding stable current value, and then measure the stable current value under each electrode potential point by point to obtain a complete polarization curve. The expression static method is ambiguous and easily misunderstood; this expression has been removed from the new manuscript, and specific steps are used to describe the operations related to polarization curve measurements:

The electrochemical test uses a three-level system, and the measurement steps are as follows: (1) Place the cable, reference electrode and platinum-plated titanium anode into the insulated column container and add the acid rain solution configured in advance; after the cable has been immersed in the acid rain solution for 2 hours, turn on the power switch of the constant potential meter, allow the instrument to warm up for 30 minutes and then set the constant potential meter to the constant potential working state. (2) Connect the working electrode of the constant potential meter to the cable, the reference electrode to the saturated glycerol reference electrode and the counter electrode to the platinum-plated titanium auxiliary anode, and press the load key and the reference key at the same time to make the instrument work in the reference state. The reference indicator lights up and the value shown in the potential display bar is the reference potential of the cable in the acid rain solution relative to the reference electrode, which is the open circuit potential of the cable. (3) Press the load and the electrolytic cell buttons at the same time to place the instrument in the electrolytic cell mode, with the open circuit potential of the cable as the initial value, adjust the voltage by rotating the DC button to the positive direction by 20mV each time, and record and observe the corresponding current and voltage data for one minute after each voltage adjustment. (4) Taking the open circuit potential of the cable as the initial value, adjust the voltage by rotating the DC button in a negative direction by 20mV each time, and record and observe the corresponding current and voltage data for one minute after each voltage adjustment. (5) Record the measured voltage and current data, and draw a polarization curve. Repeat the above steps to measure the polarization curves of the cable in different pH solutions, as shown in Figure 2.

Point 5: “few air bulbs generate”- you mean “few air bubbles are generated”? or hydrogen bubbles??

“air bulbs”- ??? perhaps are hydrogen bubbles not air.

Response 5: Your suggestion is correct. when the protection voltage is set too low, hydrogen bubbles will be generated on the cable surface and hydrogen embrittlement will occur. Hydrogen bubbles are exactly what we want to express, so we changed the air bulbs in the original manuscript to hydrogen bubbles.

Delete: In measuring, the protective current increases gradually with the its voltage becoming negative meanwhile a few air bulbs generate on the cable surface. When the protective voltage becomes more negative than -1.400Vm, lots of air bulbs generated on the cable surface.

Replace with: During the measurement, the protective current increases gradually as its voltage becomes negative; meanwhile, a few hydrogen bubbles are generated on the cable surface. When the protection voltage of the cable ICCP system is more negative than -1.400V, lots of hydrogen bubbles are generated on the cable surface.

Point 6: “Fig. 5(b)”- Can be the yellow color an evidence of a dealloying corrosion process? is there some bronze material nearby?

Response 6: Fig. 5(b) shows the red-brown corrosion products produced by the corrosion of the steel wire, which is actually the rust mixture produced by the corrosion of the iron matrix inside the wire after the consumption of the galvanized layer on the surface of the steel wire and can be used to judge the corrosion process of the steel wire to a certain extent. Through Fig. 5(b), it can be known that, under the same corrosion time, the iron matrix inside the galvanized layer of the steel wire not protected by the ICCP device has been corroded, and its corrosion degree is more severe than that of the protected steel wire.

Point 7: “no bronzing rust”- ? ? ?

“bronzing rust”- Could be from dealloying corrosion?

Response 7: Our expression is incorrect; we want to express the “red-brown rust” produced by the steel wire corrosion process. We have replaced and corrected the expression “bronzing rust” in the new manuscript.

Point 8: “protection efficiency of the galvanized coat gradually reduces”- What happens with the zinc oxide residues after the galvanized coat is exhausted? Can this material provoke some problem?

Response 8: For cables, there are two main sources of zinc oxide. Firstly, due to the active nature of zinc, the galvanized layer is the first to react with oxygen to form a dense layer of zinc oxide, which protects the cable from corrosion due to its more stable chemical properties. Secondly, despite the presence of the protective film of ZnO, the galvanized layer directly exposed to the environment will still corrode, and the zinc ions generated by the consumption of the galvanized layer will also generate a portion of ZnO hydrate by participating in the reaction with oxygen and water in the air. The process of electrochemical reaction with Zn-related substances is uniformly referred to as “galvanized layer depletion” in this study, focusing on the two stages of “galvanized layer depletion” and “iron substrate depletion”. When the galvanized layer has been completely depleted, zinc oxide may have some effect on the corrosion process and retard the development of cable corrosion, which is similar to the corrosion products produced when the iron matrix is depleted. Based on your recommendations, we have highlighted the areas of corrosion products.

Delete: This result is consistent with mean corrosion rate change trend of cables at different time period in steel wire corrosion test and indicates the wrapping and accumulation of corrosion products can delay corrosion development to a certain degree.

Replace with: This result is consistent with the mean corrosion rate change trend of the cables at different time periods in the steel wire corrosion test, and indicates that the common encapsulation and accumulation of both galvanized layer corrosion products in the SACP stage and iron matrix corrosion products in the ICCP stage can delay corrosion development to a certain degree.

Point 9: “Conclusions”- Please improve the clarity of the conclusions. They are hard to read.

Response 9: We have reorganized and discussed the conclusion section of the full text, further streamlining the original manuscript's conclusions, replacing some sentences, and using simpler and more direct expressions:

(1) The impressed current generated by a more negative protection voltage can provide more efficient corrosion protection to bridge cables. However, an excessively negative voltage may lead to hydrogen embrittlement of the steel wires inside the cable due to overprotection. The test showed that the rational range of the protection voltage is between -1.130V and -1.150V. When the protection voltage output by ICCP is more negative than the set range, the corroded steel wire will undergo brittle fracture and the fracture surface will have a radial shape. When the protection voltage output by ICCP is more positive than the set range, the corroded steel wire will undergo a tolerant fracture, showing necking down and a shear fracture ridge.

(2) The bridge cables are jointly protected by both the impressed current and cathodic protection of the sacrificial anode. The corrosion product can delay the corrosion development of the cables. The protection efficiency of the cable ICCP system increases gradually with corrosion development, and the SACP protection efficiency of the galvanized coat decreases gradually with corrosion development. The cable corrosion protection is transformed from joint protection to protection dominated by the impressed current, thus steadily inhibiting corrosion generation.

(3) Ideas for the installation of auxiliary anodes in actual bridge projects: The cables installed in the lowest section of the midspan of the main cable in suspension bridges, the cables installed in the anchorage section of cable-stayed bridges and the cables mounted in the anchorage section of the short hanger in semi-through and through bridges all are vulnerable to the impacts of fracture damage, water collection and impact effects, and serious corrosion-induced fatigue problems exist in these cables. The auxiliary anode is densely twined to provide a more adequate and uniform impressed current so as to further generate higher-efficiency protection.

Point 10: “reals the corrosion”- ? ? ?

Response 10: The complete sentence we would like to express is “reveals the corrosion protection mechanism of the ICCP system of bridge cables”, and the wrong word “reals” has been corrected in the new manuscript.

Delete: This study analyses the behavior of cable ICCP in its corrosion protection of cable and reals the corrosion protection mechanism of cable ICCP for bridges.

Replace with: This study analyzed the behavior of cable ICCP in its corrosion protection of a cable and revealed the corrosion protection mechanism of cable ICCP for bridges.

Thank you for reviewing it again.

All Authors

Reviewer 2 Report

The manuscript, entitled „ Test Study of Bridge Cable Corrosion Protection Mechanism 1 Based on Impressed Current Cathodic Protection” is relevant to the scope of this journal. It is an interesting study that can bring valuable information to specialists.

The authors have made a good synthesis of the literature that provides an overview of the research evolution in this area.

However, some points need to be addressed prior to the publication of this manuscript. My comments/suggestions are given:

1.     The authors must specify the speed with which the potentiodynamic curves presented in Figure 2 were recorded.

2.     In addition to the open circuit potentials, it would be interesting to calculate the corrosion potential values, and the corrosion current density, from which the corrosion rate can then be determined. Those established from gravimetric measurements are also instructive, but those from polarization curves are more precise. A comparison could also be made.

3.     Authors must pay attention to small mistakes such as the one in line 224! "1,400Vm".

4.      A scale introduced in images a and b from Figure 3, respectively a-d from Figure 4, would be more edifying.

Author Response

Response to Reviewer 2 Comments

Dear reviewers and editors of Lubricants, thank you for reviewing this paper and providing valuable comments during your busy schedule. Our team further optimized the structure of the manuscript, sorted out the conclusions again, and improved the language. We have carefully read and considered the issues and suggestions you pointed out. After discussing and reorganizing them together, we now explain and correct them.

Point 1: The authors must specify the speed with which the potentiodynamic curves presented in Figure 2 were recorded.

Response 1: We have supplemented the text with relevant information about polarization curves, including detailed steps for using the instrument and the speed of the dynamic potential curve. We have added the following to the new manuscript:

The electrochemical test uses a three-level system, and the measurement steps are as follows: (1) Place the cable, reference electrode and platinum-plated titanium anode into the insulated column container and add the acid rain solution configured in advance; after the cable has been immersed in the acid rain solution for 2 hours, turn on the power switch of the constant potential meter, allow the instrument to warm up for 30 minutes and then set the constant potential meter to the constant potential working state. (2) Connect the working electrode of the constant potential meter to the cable, the reference electrode to the saturated glycerol reference electrode and the counter electrode to the platinum-plated titanium auxiliary anode, and press the load key and the reference key at the same time to make the instrument work in the reference state. The reference indicator lights up and the value shown in the potential display bar is the reference potential of the cable in the acid rain solution relative to the reference electrode, which is the open circuit potential of the cable. (3) Press the load and the electrolytic cell buttons at the same time to place the instrument in the electrolytic cell mode, with the open circuit potential of the cable as the initial value, adjust the voltage by rotating the DC button to the positive direction by 20mV each time, and record and observe the corresponding current and voltage data for one minute after each voltage adjustment. (4) Taking the open circuit potential of the cable as the initial value, adjust the voltage by rotating the DC button in a negative direction by 20mV each time, and record and observe the corresponding current and voltage data for one minute after each voltage adjustment. (5) Record the measured voltage and current data, and draw a polarization curve. Repeat the above steps to measure the polarization curves of the cable in different pH solutions, as shown in Figure 2.

Point 2: In addition to the open circuit potentials, it would be interesting to calculate the corrosion potential values, and the corrosion current density, from which the corrosion rate can then be determined. Those established from gravimetric measurements are also instructive, but those from polarization curves are more precise. A comparison could also be made.

Response 2: The open circuit potential, the corrosion potential values, and the corrosion current density are indeed helpful in determining the corrosion rate. The measurements on the polarization curves in the paper are for the preliminary selection of the protection potential range of the ICCP system for the cable, and the relevant parameters obtained from the polarization curves were not initially the focus of this study. In the discussion of the protection mechanism of the ICCP system in Section 4, the analysis is also primarily based on open circuit potential as the main parameter. We mainly want to express which of the two is the key to driving corrosion protection between ICCP and SACP, and the contribution of the two in the joint cathodic corrosion protection of cables, so the qualitative relationship between open circuit potential and cable corrosion tendency and the corrosion rate is used for discussion and analysis. At the same time, we have supplemented and rewritten the discussion of open circuit potential, please check it out. For your proposal to consider the influence of parameters such as corrosion potential value and corrosion current density on the corrosion rate of cables under corrosion protection devices, we believe that these electrochemical parameters are particularly worthy of study in the corrosion protection of bridge cables, and our team will focus on this and conduct related research in the future.

Point 3: Authors must pay attention to small mistakes such as the one in line 224! “1.400Vm”.

Response 3: We have changed “1.400Vm” to “1.400V” in Line 224.

Point 4: A scale introduced in images a and b from Figure 3, respectively a-d from Figure 4, would be more edifying.

Response 4: In Fig. 4(a)-(d), we have improved the image, added the appropriate scale, and described it in the text, please check it out. In Fig. 3(a)and(b), the dimensions of the relevant cable sections are described in detail in the text, and for comparison, the length of the HDPE sheath is marked with blue segments in Fig. 3(a).

Thank you for reviewing it again.

All Authors

Reviewer 3 Report

the topic may be interesting per se, but the manuscript is extremely hard and difficult to be read. too long sentences, a lot of improper expressions, lacking of clear and well defined structure. 

Author Response

Response to Reviewer 3 Comments

Dear reviewers and editors of Lubricants, thank you for reviewing this paper and providing valuable comments during your busy schedule. Our team further optimized the structure of the manuscript, sorted out the conclusions again, and improved the language. We have carefully read and considered the issues and suggestions you pointed out. After discussing and reorganizing them together, we now explain and correct them.

Point 1: The topic may be interesting per se, but the manuscript is extremely hard and difficult to be read. too long sentences, a lot of improper expressions, lacking of clear and well defined structure.

Response 1: Our team optimized the article's language and structure by breaking down some of the long sentences into more fluent, simple sentences, correcting individual incorrect words and phrases, improving some inappropriate expressions, and making the manuscript easier to read. The discussion of the mechanism and conclusion were redescribed, and the content of the article was fully corrected and supplemented according to the opinions of various experts.

We have corrected and enhanced the full text, and the article has been touched up by professionals. The revised text has been uploaded for your review.

Thank you for reviewing it again.

All Authors

Reviewer 4 Report

The manuscript provides relatively new and interesting results on cathodic protection  of bridge cables. Howver, some improvement of the manuscript is needed.

There are several gramatic errors that should be corrected.

Authors describe sacrificial anode cathodic protection but it should be more clearly explaind that this is actually galvanic protection by coating.

Line 206- what would be static method? Please provide information on instruments and potential range used in polarization studies.

Why were studies conducted at 10°C?

From polarization curves it is clear that in higher pH solutions corrosion is not accelerated, as stated in the text, but actually decreased. Please correct the explanation.

It is not clear why pH3 improves protecting efficiency? It is higly unlikely that such low pH will be observed in practice?

-line 224- what does it mean that the potential is more negative? Compared to what?

271- what would be electronic tester?

280- please rewrite "emphatically considered"it is not clear what would that mean?

307-"more dense change gradient" please explain what do you mean by that

There is no description of experimental conditions for tensile strenght test

Author Response

Response to Reviewer 4 Comments

Dear reviewers and editors of Lubricants, thank you for reviewing this paper and providing valuable comments during your busy schedule. Our team further optimized the structure of the manuscript, sorted out the conclusions again, and improved the language. We have carefully read and considered the issues and suggestions you pointed out. After discussing and reorganizing them together, we now explain and correct them.

Point 1: Authors describe sacrificial anode cathodic protection but it should be more clearly explaind that this is actually galvanic protection by coating.

Response 1: The cathodic protection of the sacrificial anode is indeed current protection through the galvanized coating on the cable surface. In original manuscript, we used the phrase “The SACP uses zinc as the galvanized coat which has a lower potential than that of cable steel wire to provide protection current to the iron matrix” to express that the cathodic protection of the sacrificial anode is performed using the coating. We did not give a sufficient explanation in the original manuscript, and now, based on your comments, we have further emphasized and explained the protection mechanism in the new manuscript.

Delete: The SACP uses zinc as the galvanized coat which has lower potential than that of cable steel wire to provide protection current to the iron matrix and uses current produced by anode solution to protect cathodic surface.

Replace with: The SACP uses zinc as the galvanized coat, which has a lower potential than the cable steel wire to provide a protection current to the iron matrix. It uses the elemental zinc coating as the anode and protects the cathode surface using the current generated by the dissolution of the anode.

Point 2: Line 206- what would be static method? Please provide information on instruments and potential range used in polarization studies.

Response 2: What the static method is trying to convey is that keep the electrode potential at a certain value, measure the corresponding stable current value, and then measure the stable current value under each electrode potential point by point to obtain a complete polarization curve. The expression “static method” is ambiguous and easily misunderstood; this expression has been removed from the new manuscript, and specific steps are used to describe the operations related to polarization curve measurements.

Based on your comments, the following information about the potential range, instruments, and their usage in polarization studies have been added to the new manuscript:

The electrochemical test uses a three-level system, and the measurement steps are as follows: (1) Place the cable, reference electrode and platinum-plated titanium anode into the insulated column container and add the acid rain solution configured in advance; after the cable has been immersed in the acid rain solution for 2 hours, turn on the power switch of the constant potential meter, allow the instrument to warm up for 30 minutes and then set the constant potential meter to the constant potential working state. (2) Connect the working electrode of the constant potential meter to the cable, the reference electrode to the saturated glycerol reference electrode and the counter electrode to the platinum-plated titanium auxiliary anode, and press the load key and the reference key at the same time to make the instrument work in the reference state. The reference indicator lights up and the value shown in the potential display bar is the reference potential of the cable in the acid rain solution relative to the reference electrode, which is the open circuit potential of the cable. (3) Press the load and the electrolytic cell buttons at the same time to place the instrument in the electrolytic cell mode, with the open circuit potential of the cable as the initial value, adjust the voltage by rotating the DC button to the positive direction by 20mV each time, and record and observe the corresponding current and voltage data for one minute after each voltage adjustment. (4) Taking the open circuit potential of the cable as the initial value, adjust the voltage by rotating the DC button in a negative direction by 20mV each time, and record and observe the corresponding current and voltage data for one minute after each voltage adjustment. (5) Record the measured voltage and current data, and draw a polarization curve. Repeat the above steps to measure the polarization curves of the cable in different pH solutions, as shown in Figure 2.

Point 3: Why were studies conducted at 10°C?

Response 3: Sorry, due to our negligence, there was an error in the writing of the units, the corrosion test and the measurement of the polarization curve were carried out at 35 degrees, and the temperature conditions for the test have been supplemented to the text. The temperature of 35 degrees was chosen as the test condition because it is a temperature environment with a high corrosion rate for the cable, which is in line with the idea of accelerated testing. Relevant content and literature have been added to the manuscript, and the erroneous writing has been corrected and deleted:

It is worth noting that when corrosion of a specimen occurs in an acidic solution, an ambient temperature of about 35°C will accelerate its corrosion rate [46, 47]. Therefore, in order to accelerate the efficiency of the test and verify the feasibility of the ICCP system, the test was conducted in the environmental test chamber of the State Key Laboratory of Mountain Bridge and Tunnel Engineering, Chongqing Jiaotong University, with the ambient temperature controlled at 35°C and the temperature error controlled within 2°C.

Point 4: From polarization curves it is clear that in higher pH solutions corrosion is not accelerated, as stated in the text, but actually decreased. Please correct the explanation.

Response 4: In this study, the polarization curves of the cables at three pH values were measured. From the polarization curves, it can be seen that although the corrosion rate difference between the three polarization curves is not obvious, the cables have a more negative open-circuit potential in more acidic solutions, and their corrosion susceptibility and corrosion rates are slightly higher. We fine-tuned the color of the polarization curves to make this phenomenon appear more obvious and made some changes to the description of the text in the new manuscript.

Point 5: It is not clear why pH3 improves protecting efficiency? It is higly unlikely that such low pH will be observed in practice?

Response 5: For your first question, the original manuscript wants to express that the use of pH=3 solution for corrosion can accelerate the corrosion of the cable to better demonstrate the protection efficiency of the ICCP system. Because this expression is indeed easy to misunderstand for the readers, we have corrected it in the corresponding position in the new manuscript.

Delete: However, the cable in solvent with higher pH displays accelerated corrosion speed. Hence the acid rain solvent of value pH=3 selected can improve efficiency and highlight the results of the test.

Replace with: The cable in the solvent with a higher pH displayed an accelerated corrosion speed; hence, the selected acid rain solvent with pH=3 can improve the efficiency of the ICCP corrosion protection test and highlight the results of the test.

For your second question, it is indeed difficult to encounter such a low pH in the actual project. The reason why this study chose a strongly acidic solution with a pH of 3 is that it can accelerate the corrosion of the cable and judge whether the ICCP system designed by this institute can effectively protect the cable in a short period of time and how effectively it is to protect the cable, which can provide some reference value for the actual project.

Point 6: Line 224- what does it mean that the potential is more negative? Compared to what?

Response 6: In the original manuscript, we wanted to express that when we set a voltage more negative than -1.400mV (e.g. -1.500mV, -1.600mV) to the cable using the ICCP system, a large number of hydrogen bubbles are generated on the surface of the cable. We did not express this expression clearly, and it has now been corrected in the new manuscript.

Delete: When the protective voltage becomes more negative than -1.400V.

Replace with: When the protection voltage of the cable ICCP system is more negative than -1.400V.

Point 7: Line 271- what would be electronic tester?

Response 7: Electronic tester refers to an electronic pH tester, and the original manuscript was intended to mean the determination of the pH of a solution using an electronic pH tester, which has now been corrected.

Delete: electronic tester

Replace with: electronic pH tester

Point 8: Line 280- please rewrite “emphatically considered” it is not clear what would that mean?

Response 8: What we originally wanted to express was: the corrosion protection device of the cable mainly plays a role in the initial stage of cable corrosion, because when the cable corrosion protection device is seriously broken or the degree of corrosion is more severe, the corrosion process becomes less easy to control. In the actual project to block the corrosion of the cable in these two stages, so we mainly consider the two stages in the early stage of cable corrosion: the consumption of the galvanized layer and the initial participation reaction of the iron matrix. We set the cable corrosion timer to 336 hours, during which time the accelerated corrosion test fully simulates the pre-corrosion stage of the cable under service conditions and verifies the effectiveness of the ICCP device. We have improved this area to express our meaning more clearly.

Delete: The corrosion protection of bridge cables particularly is subject to impacts of dissipation of galvanized coating and iron matrix preliminary participation in reaction. Hence early behavior of corrosion development of cable is emphatically considered and each test lasts for 336h.

Replace with: The corrosion protection of bridge cables, in particular, is subject to the impacts of the dissipation of the galvanized coating and iron matrix’s preliminary participation in the reaction, which is the initial stage of cable corrosion development. In order to focus on the early behavior of cable corrosion development, the corrosion test time was set to 336h.

Point 9: Line 307- “more dense change gradient” please explain what do you mean by that.

Response 9: In Section 3.1, we set 4 operating conditions for the ICCP system of the cable: no voltage protection set, -1.070 V voltage protection, -1.150 V voltage protection, and -1.250 V voltage protection. In Section 3.2, in order to study the protective effect of different impressed currents on the steel wire more extensively, we use the ICCP system to set 6 working conditions for the steel wire: no voltage protection (118h), no voltage protection (336h), -1.070V voltage protection, -1.130V voltage protection, -1.200V voltage protection, and -1.300V voltage protection. Under different operating conditions, the protection voltage interval of the steel wire is smaller, so we use “more dense change gradient” to describe it. We have improved this statement to give the reader a clearer idea of what we are thinking.

Delete: In the test more dense change gradient of the protective voltage is set to the steel wire and each test steel wire group consisting 4 steel wires are protected by voltages of -1.070V, -1.130V, -1.200V and -1.300V, respectively.

Replace with: In the test, a denser change interval of the protective voltage was set for the steel wires, and the test steel wire groups consisting of four steel wires were protected by voltages of -1.070V, -1.130V, -1.200V and -1.300V.

Point 10: There is no description of experimental conditions for tensile strenght test.

Response 10: In the new manuscript, we have added the following to section 3.2.2 to add a description of the tensile strength test of the steel wire:

A computer-controlled twin-arm-type electronic universal testing machine equipped with the test analysis software SANS-Power Test DOOC from the structural laboratory of Chongqing Jiaotong University was used to test the mechanical properties of the corroded steel wires. Equipment model: WDW-50; equipment specifications: 5t; equipment accuracy level: 0.5 level. The equipment can complete the tensile, compression, bending, shear and basic static mechanical tests of composite materials as well as tear, friction and cycle tests. In order to further determine and set the protective range of the voltage in the ICCP system for the steel wires, the four corroded steel wires protected by different voltages were uniaxially tension-tested with the universal testing machine, and the stretching loading rate was 3mm/min. The steel wire was tensioned until it broke, determining whether the steel wire suffered from hydrogen embrittlement due to a low protection potential setting by stretching the steel wire fracture shape, and thus determining the maximum protection voltage for the steel wire, as shown in Figure 6.

Thank you for reviewing it again.

All Authors

Round 2

Reviewer 3 Report

accept